# Investigation of phytochemical and biochemical attributes of hypoglycemic activity of *Atriplex crassifolia* (C.A. Mey) extracts in alloxan diabetic animal model

Sarah Rehman[1,2], Saiqa Ishtiaq[1,3]*, Sairah Hafeez Kamran[4]*,
Muhammad Khalil-ur-Rehman[1], Syeda Farheen Fatima[1], Numera Arshad[5], Saira Rehman[6]

**1** Punjab University College of Pharmacy, University of the Punjab, Lahore, Pakistan, **2** Faculty of Pharmacy, Salim Habib University, Karachi, Pakistan, **3** Center for the Study of Human Health, Emory College of Art and Science, Emory University, Atlanta, Georgia, United States of America, **4** Department of Pharmacology, Institute of Pharmacy, Faculty of Pharmaceutical and Allied Health Sciences, Lahore College for Women University, Lahore, Pakistan, **5** Department of Pharmacy, COMSATS University Islamabad Lahore-Campus, Lahore, Pakistan, **6** Faculty of Pharmaceutical Sciences, Lahore University of Biological and Applied Sciences, Lahore, Pakistan

* sairah.hafeez@lcwu.edu.pk (SHK); saiqa.pharmacy@pu.edu.pk (SI)

## Abstract

Diabetes mellitus is a widespread medical and public health issue that can lead to significant health impairment. The objective of the study was to assess the antidiabetic effectiveness of various extracts of *Atriplex crassifolia* (*A. crassifolia*), namely n-hexane (ACNH), dichloromethane (ACDCM), ethyl acetate (ACEA), and n-butanol (ACNB). The initial qualitative analysis revealed the presence of bioactive components in all extracts, such as alkaloids, glycosides, phenols, flavonoids, terpenes, and saponins. ACEA and ACNB exhibited considerable radical scavenging activity at different doses, as determined by the DPPH free radical scavenging experiment used to assess *in-vitro* antioxidant activity. Male rats were administered 150 mg/kg of alloxan monohydrate to develop diabetes. A preliminary investigation was conducted to evaluate the potential of all extracts to lower blood glucose levels using an oral glucose tolerance test (OGTT) at three different doses (250, 500, and 1000 mg/kg/day). ACEA and ACNB exhibited promising results in the OGTT, thus prompting a 28-day hypoglycemic research to confirm their biological effectiveness in treating diabetes. When taken orally, ACEA at doses of 500 and 1000 mg/kg/day, and ACNB at doses of 250, 500, and 1000 mg/kg/day, effectively recovered the concentrations of urea, creatinine, glucose, cholesterol, triglycerides, HDL, and liver enzymes (AST, ALT, and ALP). The histopathological evidence corroborated the conclusions. HPLC and GC-MS were employed to identify the presence of medically important phytoconstituents in ACEA and ACNB. Myrecitin and Quercitin were found using High Performance Liquid Chromatography (HPLC) in ACEA, while sinapic acid was identified in ACNB. The GC-MS analyses of ACEA and ACNB revealed the presence of several

**Data availability statement:** All relevant data are within the manuscript and its Supporting information files.

**Funding:** The author(s) received no specific funding for this work.

**Competing interests:** The authors have declared that no competing interests exist.

compounds with hypolipidemic, antioxidant, and anti-diabetic properties. The present study suggests that *A. crassifolia* has potential as a natural remedy for managing hyperglycemia, diabetic hyperlipidemia, and other diabetes-related problems.

## Introduction

Diabetes mellitus (DM) is a widespread metabolic disorder that has great impact on people's general health as well as their social and economic well-being. It has been projected that by 2030, the global prevalence of the disease would increase to 552 million individuals [1]. This disease is characterized by an increase in blood glucose level because of a decrease in either the production (Type I DM) or the action of insulin (Type II DM). Uncontrolled DM leads to the dysfunction or complete destruction of several organs, including the neurological system, the blood vessels in the retina, and the kidneys [2].

Currently DM is managed by insulin and oral hypoglycemic agents including sulfonylureas, biguanides, PPAR's etc. Long term administration of these drugs may cause undesired effects and reduce the quality of life in diabetic patients. Many scientists have scientifically proven that herbal products like cinnamon, turmeric, garlic possess promising antidiabetic effects. Traditional medicine is easily accessible due to its low cost and are preferred by people with low socioeconomic status in the developing world [3]. The antidiabetic mechanisms of bioactive secondary metabolites derived from traditional herbal products primarily encompass insulinotropic activity, reduced insulin resistance, increased glycogen storage in the liver and skeletal muscle, inhibition of monosaccharide absorption, and glucose production, along with an added benefit of anti-inflammatory and antioxidant properties [4].

Herbal medicine offers a safer and more effective method for disease management. A range of bioactive chemicals found in medicinal plants affect certain pathways of the pathogenic mechanisms associated with diabetes and its consequences. The genus Atriplex consists of over 300 halophyte species and is extensively dispersed across various regions of North America, Asia, and Australia [5]. Recent investigations indicate that extracts from Atriplex species exhibit a range of biological activities, including antioxidant, anticholinesterase, antibacterial, antifungal, and antiparasitic properties. In-vivo studies have demonstrated nephroprotective, hepatoprotective, and anti-diabetic properties. This genus encompasses a diverse array of phytoconstituents, including flavonoids, sterols, phytoecdysteroids, saponins, and alkaloids. Plants of this genus are believed to be rich in vanillin, kaempferol, ferulic acid, and other therapeutically valuable phytoconstituents [6]. *Atriplex crassifolia* (C.A. Mey), an edible plant of the Chenopodiaceae family, with a notable historical significance in Pakistan as a treatment for infectious and inflammatory ailments. The leaves of *A. crassifolia* are documented to be efficacious against liver and various infectious disorders and are utilized for treating throat infections and jaundice in certain regions of Pakistan [7]. The entire plant is burned to produce ash, which is combined with a little amount of sesame oil and applied externally to alleviate rheumatic discomfort. Due to the little scientific data concerning the phytochemistry and

biological significance of *A. crassifolia*, the present study seeks to discover the phytochemical components, in vitro antioxidant properties, and in vivo antidiabetic potential of the plant.

## Materials and methods

The plant *Atriplex crassifolia* was collected from District Kasur and was authenticated by Dr. Zaheer-ud-Din, Curator of Dr. Sultan Herbarium GC University, Lahore against a voucher specimen number **GC.Herb.Bot.3618.** The plant was washed with distilled water, shade dried, pulverized and stored in air-tight container for further use.

### Drugs and chemicals

The standard drug, glibenclamide was procured from Sanofi Aventis, Pakistan. Alloxan monohydrate, myricetin, quercetin, kaempferol, sinapic acid, caffeic acid, DPPH (2,2-Diphenyl 1- picryl-Hydrazyl), ascorbic acid and solvents like n-hexane, dichloromethane, ethyl acetate and n-butanol were bought from Merck. Germany. Glucometer used to check blood glucose levels was ACCU-Check Roche Germany. All chemicals were of analytical grade.

### Preparation of extracts of *A. crassifolia*

Successive solvent extraction of the plant was carried out following the procedure of Solikhah et al [8] using solvents of ascending polarity (n-hexane, dichloromethane, ethyl acetate and n-butanol) through cold maceration method for 72 hours with intermittent shaking. All the extracts were evaporated using rotary evaporator under relegated pressure at 60 rpm and 40°C to obtain concentrated extracts. The resultant extracts were placed in glass beakers and stored in refrigerator to prevent contamination. The extracts were designated as *A. crassifolia* n-hexane extract (ACNH), *A. crassifolia* dichloromethane extract (ACDCM), *A. crassifolia* ethyl acetate extract (ACEA) and *A. crassifolia* n-butanol extract (ACNB).

### Phytochemical analysis

The preliminary phytochemical tests were performed to screen the presence of alkaloids, glycosides, carbohydrate, flavonoids, phenols and other phytoconstituents in ACNH, ACDCM, ACEA and ACNB [9].

### *In vitro* antioxidant analysis with DPPH (2,2-Diphenyl 1- picryl-Hydrazyl) assay

The antioxidant activity of ACNH, ACDCM, ACEA and ACNB was corroborated on the basis of their ability to scavenge stable 1, 1- diphenyl 2-picrylhydrazyl (DPPH) free radical by comparison with known anti-oxidant, ascorbic acid, employing the method described by Ishtiaq et al [10] with slight modifications. 3 mL of methanolic solution of DPPH (0.1 mmol/L) was mixed in different concentrations (15, 30, 60, 125, 250, 500, 750 & 1000 µg/ml) of plant extracts. The mixtures were shaken vigorously and incubated in dark at room temperature for one hour. Absorbance was measured at 517 nm using UV spectrophotometer (Shimadzu 1650) keeping methanol as blank. Reaction mixtures demonstrating more discoloration of DPPH and a lower absorbance represented higher free radical scavenging activity. Percentage inhibition for each dilution was calculated by using the following formula.

$$\% \text{ Inhibition } = \text{ A control } - \text{ B sample } / \text{ A control } \times 100$$

Where A represents the absorbance of the control, B represents the absorbance of sample. The free radical scavenging activity of the plant extracts were expressed as IC50.

### GC-MS analysis of *A. crassifolia* ethyl acetate (ACEA) and *A. crassifolia* n-butanol extract (ACNB)

ACEA and ACNB extracts were subjected to GC-MS analysis. Agilent-B-7890 having 5977-B mass spectrometer detector was used for analysis. The stationary phase comprised of 100% dimethyl polysiloxane (Dimensions: L-30 m, D-0.25 mm,

pore size 0.25 µm). Injector temperature was set at 250 ˚C and that of the inter-phase at 280 ˚C. The equipment was set at scan mode with −70 ev ionization energy and the temperature of ion source was kept 230 ˚C. Helium at a flow rate of 1 ml/minute was selected as carrier gas. The analysis time was 60 minutes [11].

## High performance liquid chromatography (HPLC) of *A. crassifolia* ethyl acetate (ACEA) and *A. crassifolia* n-butanol extract (ACNB)

ACEA and ACNB were analyzed for the presence of flavonoids (myrecitin, quercitin, kaempferol) and phenolic acids (sinapic acid, caffeic acid, gallic acid) by using High performance Liquid chromatography (model LC-10A Shimadzu, Kyoto, Japan) containing Rheodyne injector, 2 LC-10 AS pumps, SCL- 10A system control unit, ODS reverse phase (C18) column and CTO-10 A column oven. Data was analyzed using LC-10 software (Data acquisition class). 20 µL of sample was filtered and injected into column (Particle size: 5 µm, 250 mm). Mobile phase constituted of a 50:50 v/v mixture of 3% trifluoro acetic acid (Solvent A) and 80:20 v/v acetonitrile and methanol (Solvent B). The flow rate was adjusted at 1 ml/min. The column was controlled thermostatically at 30˚C. HPLC chromatograms were detected using SPD-10A UV-visible detector at three different wavelengths (272, 280 and 310 nm), according to the absorption maxima of analyzed compounds. Each compound was recognized by its retention time and by empaling with standards under the same conditions [12].

## Animals

Healthy male Wistar rats (180–250 g) were housed in polypropylene cage at University College of Pharmacy, University of the Punjab (Lahore, Pakistan). Ethical approval to conduct animal study was secured from Institutional Ethics Review Board (No. D/208/FIMS). Rats were provided commercial rodent diet and kept on 12 h light: 12 h dark cycle, 50% humidity and 28 ± 2˚C.

## Induction of experimental diabetes

Chemical diabetes was induced in the animals with 10% solution of Alloxan monohydrate prepared in distilled water and injected intraperitoneally at dose of 150 mg/kg. Rats were then catered with bottles containing 5% w/v solution of glucose in their cages for next 24 hours to prevent hypoglycemia encountered after alloxan administration. Glucose levels were tested after 48–72 hours. Animals having blood glucose levels greater than 200 mg/dL after 72 hours of alloxan administration were selected for the study [13].

## Experimental design

**Oral glucose tolerance test (OGTT) of diabetic rats.** An evaluation of oral glucose tolerance was conducted on a group of seventy rodents that had been induced with diabetes, as previously mentioned. The animals that were chosen underwent a 24-hour fast, during which they were provided with unrestricted access to water. Following this, the animals were randomized into fourteen groups of five rodents each. The digital glucometer was employed in conjunction with the tail tilting method to ascertain the basal blood glucose levels (BGL). Following that, oral administration of varying concentrations (250, 500, 1000 mg/kg) of the aforementioned extracts (ACNH, ACDCM, ACEA, and ACNB) was conducted on 12 groups of diabetics. The diabetic and standard control groups (glibenclamide 10 mg/kg) were the thirteenth and fourteenth, respectively. The 15th group, which served as the normal control, was comprised of normal animals. Blood glucose levels were reassessed (baseline) thirty minutes after extract administration; thereafter, an oral glucose dose of 2g/Kg) was administered to all animals. Blood glucose levels were measured 30, 60, 90, and 120 minutes after glucose administration [14].

**Hypoglycemic study model.** In a subsequent 28-day hypoglycemic study design, the plant extracts that demonstrated exceptional hypoglycemic potential in the preliminary oral glucose tolerance test (OGTT) were evaluated.

Forty diabetic rats were allocated at random into eight groups (n = 5). Group 1 served as normal control, Group 2; Diabetic control, Group 3: glibenclamide (10 mg/kg), Group 4–6: ACEA 250, 500 and 1000 mg/kg, Group 7–9: ACNB 250, 500 and 1000 mg/kg respectively. The blood glucose levels (BGL) were assessed using a glucometer (tail tilting method) from the third day until the 28th day of the study. The standard drug and extracts were administered via oral gavage.

**Evaluation of weight and biochemical parameters.** The rats' weights were measured at the beginning of the experiment, as well as on the 15th and 28th days of the study. The overall well-being of the animals was observed and assessed throughout the duration of the study. On the final day of the experiment, the animals were given anesthesia and blood samples were obtained by puncturing the heart to measure the levels of serum cholesterol, triglycerides, LDL, HDL, and vLDL, as well as urea, creatinine, AST, ALT, and ALP. The values were determined spectrophotometrically using kits provided by Diagnostic systems GmbH (Holzheim, Germany) [15].

**Histopathological analysis of pancreas of various experimental groups.** On the last day of the study, rats were anesthetized, and the pancreas were removed and preserved in 10% freshly prepared formaldehyde solution. The tissues were processed in auto Technicon and sections of 5μ thickness were mounted and stained with H&E (hematoxylin & eosin) [16].

**Statistical analysis.** All the data was evaluated using Graph Prism version 8.0.1. The results were expressed in terms of standard error of mean (SEM). The values were analyzed by One-way analysis of variance (ANOVA)followed by Dunnet test as post hoc. The values were estimated to be statistically significant at 95% confidence level. ($p < 0.05$)

## Results and discussion

### Phytochemical screening

The phytochemical analysis of several extracts obtained from *A. crassifolia* revealed the existence of many phytoconstituents, including alkaloids, glycosides, carbohydrates, phenols, flavonoids, tannins, terpenoids, lipids, and saponins (Table 1). The therapeutic efficacy of medicinal plants is attributed to the existence of bioactive constituents such as alkaloids, glycosides, flavonoids, and triterpenes. Throughout history, there have been reports of the ability of extracts derived from medicinal plants to counteract metabolic dysfunction, thereby potentially slowing down the advancement of diabetes. This beneficial effect is attributed to the existence of bioactive elements within these extracts. Our findings are consistent with

**Table 1. Results of preliminary phytochemical screening of *Atriplex crassifolia* C.A.Mey.**

| Phytochemical Groups | ACNH | ACDCM | ACEA | ACNB | ACM |
|---|---|---|---|---|---|
| Alkaloids | + | + | ++ | ++ | ++ |
| Glycosides | + | ++ | ++ | + | + |
| Carbohydrates | + | + | − | ++ | − |
| Phenols | + | + | ++ | ++ | ++ |
| Flavonoids | − | + | ++ | ++ | ++ |
| Proteins | − | − | − | + | + |
| Amino acids | − | − | − | + | + |
| Tannins | ++ | + | + | − | ++ |
| Starch | + | + | − | + | − |
| Terpenoids | + | ++ | ++ | ++ | ++ |
| Fats and Fixed oils | ++ | ++ | ++ | ++ | ++ |
| Saponins | − | − | − | + | + |

++ = Appreciable amount, + = Moderate amount, - = Not detected. ACNH: *Atriplex crassifolia* n hexane extract, ACDCM: *Atriplex crassifolia* Dichloromethane extract, ACEA: *Atriplex crassifolia* Ethyl acetate extract, ACNB: *Atriplex crassifolia* n-butanol extract.

prior research conducted on various Atriplex species, which have demonstrated the existence of comparable phytochemicals [4].

## In vitro antioxidant activity

**DPPH (2,2-Diphenyl 1- picryl-Hydrazyl) assay.** The DPPH free radical scavenging assay is considered a reliable method for evaluating antioxidant capability. The generation of a prominent deep purple colour is ascribed to the electron-donating properties of the DPPH radical. The color change of DPPH and its subsequent decolorization can be objectively measured when the antioxidant molecule accepts electrons [17]. In the current investigation, ascorbic acid served as the reference standard. ACEA and ACNB exhibited the most significant radical scavenging action among the evaluated substances at various dosages, equivalent to the inhibitory impact of ascorbic acid. The extracts demonstrated a dose-dependent radical scavenging activity, indicating that antioxidant activity increased with higher extract concentrations (Table 2). Reports indicate that elevated concentrations correlate with enhanced antioxidant activity due to an increased rate of radical scavenging. The dose-dependent enhancement in antioxidant potential may be ascribed to an increased availability of antioxidant molecules capable of scavenging free radicals at elevated concentrations [18]. The antioxidant capacity is mostly ascribed to the presence of phenols, flavonoids, terpenes, and tannins. Phenols exhibit antioxidant activity that is dependent on concentration following absorption from the gastrointestinal system. The application of high-performance liquid chromatography (HPLC) has enabled the identification and quantification of phenolic compounds in A. crassifolia. The research indicates that the structural chemistry of phenols is very conducive to free radical scavenging activity. The antioxidant capabilities of flavonoids, both primary and secondary, have been evidenced by their capacity to diminish lipid peroxidation. These substances may function as regulators for both pro-oxidant and antioxidant enzymes [19]. The GC-MS analysis revealed the presence of several antioxidant chemicals in ACEA, including phytol, tyrosol, benzophenone, methyl ethyl and butyl esters of hexadecenoic acid, n-hexadecanoic acid, cetene, linoleic acid, and

**Table 2. Percentage scavenging activity of different extracts of *A. crassifolia*.**

| Conc (μg/ml) | %age inhibition Ascorbic acid | %age inhibition ACNH | %age inhibition ACDCM | %age inhibition ACEA | %age inhibition ACNB |
|---|---|---|---|---|---|
| 15 | 35.33± 1.21 | 27.45± 0.03*** | 34.14± 0.22ns | 33.44± 0.29ns | 34.44± 0.14ns |
| 30 | 38.13± 1.49 | 24.52± 0.01*** | 34.48± 0.31* | 38.13± 0.30ns | 33.66± 0.15* |
| 60 | 39.66± 0.77 | 32.42± 0.08** | 34.76± 0.54* | 39.53± 0.70ns | 46.42± 0.31ns |
| 125 | 45.30± 0.66 | 33.69± 0.20*** | 35.53± 0.32*** | 44.30± 0.56ns | 42.66± 1.00ns |
| 250 | 65.77± 0.05 | 37.54± 0.24*** | 36.13± 0.51*** | 61.16± 0.54ns | 60.44± 0.35ns |
| 500 | 76.56± 0.57 | 41.51± 0.01*** | 43.81± 0.16*** | 71.36± 0.72ns | 81.40± 0.73ns |
| 750 | 85.26± 0.73 | 43.57± 0.38*** | 44.71± 0.06*** | 85.27± 0.45ns | 85.40± 0.65ns |
| 1000 | 95.33± 0.57 | 44.19± 0.08*** | 45.88± 0.31*** | 91.63± 0.20ns | 93.51± 0.39ns |
| IC 50 | 178.6 | 1166*** | 1114*** | 206.06ns | 183.19ns |

Results are shown as mean±SEM. *=<0.01, **=<0.001, ***=<0.0001 respectively by one way ANOVA followed by Dunnet's multiple comparison test
ACNH: *A. crassifolia* n-hexane extract, ACDCM: *A. crassifolia* dichloromethane extract, ACEA: *A. crassifolia* ethyl acetate extract, ACNB: *A. crassifolia* n-butanol extract. IC50 value is determined by preparing the calibration curve for Ascorbic acid.

vitamin E. The ACNB has been shown to contain many antioxidants, such as benzenedicarboxylic acid, hexadecenoic acid, 1-octadecene, 10-octadecenoic acid methyl ester, 1-heptadecene, and erucic acid. The ACEA and ACNB compounds exhibited a notable antioxidant capacity, as seen by their substantial percent free radical scavenging activity presented in Table 2. This finding further supports the notion that *A.crassifolia* contains bioactive chemicals that hold promise as possible therapeutic agents for various disorders.

## Chemical characterization of active extracts by GC-MS analysis

The chemical components identified through GC-MS analysis in ACEA and ACNB are presented in Tables 3 and 4. The analysis of active constituents in the extracts validated the presence of many antioxidant, anti-inflammatory, hypocholesterolemic, and antidiabetic compounds that may elucidate the plant's therapeutic properties. Prominent antidiabetic compounds identified in ACEA and ACNB include γ-sitosterol, β-sitosterol, Phytol, and 1,2-Benzisothiazole. γ-sitosterol facilitates the inhibition of gluconeogenesis and augments insulin secretion. Moreover, it reduces blood cholesterol, LDL, v-LDL, and triglyceride levels while concurrently elevating HDL levels [20]. β-sitosterol impedes glycation and promotes the regeneration of cells that produce antioxidant enzymes. This subsequently restores the impaired extracellular matrix proteins in hyperglycemic circumstances and facilitates cellular proliferation [21] The terpenes identified in ACEA (Fig 1) include phytol, cyclohexanol, and 3-ethenyl-3-methyl-2-(1-methylethenyl)-6-(1-methylethyl)-, [1R-(1. α,2. α,3. beta.,6. α)]. The prominent compounds found in ANCB were neophytadiene, α-Tocospiro B, bicyclo [3.1.1] heptane, 2,6,6-trimethyl-, [1R-(1. α,2. ß,5. ß)]-, flexibilide, and phytol (Fig 2).

Phytol is a diterpene classified as an unsaturated long-chain acyclic alcohol. It achieves its antioxidant function by eliminating reactive oxygen and nitrogen species (ROS and RNS) that are produced in the body due to cellular stress [22]. Terpenes have been documented to have antioxidant, antidiabetic, and hypolipidemic characteristics. They decrease the activity of alpha-glucosidase and promote the stability of insulin secretion [23]. Tyrosol and 2,4-Di-tert-butylphenol in ACEA, while d-α-tocopherol in ACNB demonstrate a hypoglycemic impact by inhibiting alpha glucosidase and alpha amylase [24]. The compounds 1,2-Benzisothiazole, 3-(hexahydro-1H-azepin-1-yl)-, 1,1-dioxide, found in both ACEA and ACNB extracts, belong to a group of compounds called benzothiazoles. These compounds are known to have antidiabetic effects. Benzothiazole derivatives activate PPARγ (Peroxisome Proliferator-Activated Receptor Gamma) receptors resulting in improved insulin sensitivity, regulation of blood glucose level lipid metabolism [25]. Guaifenesin in ACNB has been reported to have a preventive effect on diabetic neuropathy [26]. Additional biologically active compounds comprise of fatty acid esters, specifically hexadecanoic acid methyl ester, hexadecanoic acid ethyl ester, n-hexadecanoic acid butyl ester, and Linoleic acid ethyl ester. These compounds exhibit strong antioxidant properties and have been found to have notable effects in reducing blood sugar and lipid levels [27].

## HPLC analysis of active extracts of *A. crassifolia*

The flavonoids detected in ACEA were myricetin and quercetin, as illustrated in Fig 3. Sinapic acid and caffeic acid were identified in ACNB, as shown in Fig 4.

Flavonoids are secondary metabolites that are biologically active and polyphenolic in nature. They have a diverse array of health advantages. They have a crucial function in regulating oxidative stress by counteracting the reactive oxygen and nitrogen molecules within the body. Consequently, the extracts were examined for the existence of flavonoids and phenols through the utilization of the HPLC. Myricetin and Quercetin are widely dispersed flavonoids found in various therapeutic plants. They have a substantial antidiabetic impact through many pathways [28]. Quercetin hampers cell growth by impeding protein kinases. It reduces lipid peroxidation, which leads to the regeneration of beta cells in the pancreas, hence stimulating the release of insulin and restoring increased blood glucose levels to normal [29]. Additionally, it reduces the uptake of glucose in the intestines by blocking GLUT2. Myricetin activates the opioid receptors and stimulates the production of endogenous beta-endorphins, thereby improving insulin sensitivity and reducing blood

**Table 3. Components detected in ACEA by GC-MS analysis.**

| RT (min) | Compound name | Compound class | Molecular formula | Molecular mass | Area | Biological activity |
|---|---|---|---|---|---|---|
| 4.649 | Benzene, 1,3,5-trimethoxy- | Hydrocarbon | $C_9H_{12}$ | 120.19 | 0.13 | Anti-bacterial [43] |
| 5.785 | 2,4-Di-tert-butylphenol | Phenol | $C_{14}H_{22}O$ | 206.32 | 0.22 | Anti-diabetic [44] |
| 6.377 | Butanedioic acid, dibutyl ester | Ester | $C_{12}H_{22}O_4$ | 230.3 | 0.14 | Anti-microbial [44] |
| 6.661 | Tyrosol | Phenol | $C_{10}H_{12}O_3$ | 180.20 | 0.25 | Antioxidant, antidiabetic [45] |
| 7.193 | Benzophenone | Ketone | $C_{13}H_{10}O$ | 182.21 | 0.07 | Antioxidant, antimicrobial [46] |
| 7.658 | Cyclohexanol, 3-ethenyl-3-methyl-2-(1-methylethenyl)-6-(1-methylethyl)-, [1R-(1. α,2. α,3. beta .,6. α)]- | Terpenoid | $C_{15}H_{26}O$ | 222.36 | 0.06 | Antioxidant, anti-cancer [47] |
| 9.483 | Neophytadiene | diterpene | $C_{20}H_{38}$ | 278.5 | 0.63 | Anti-diabetic, antioxidant [48] |
| 9.561 | Pentadecanone, 6,10,14-trimethyl 2- | ketone | $C_{18}H_{36}O$ | 268.5 | 0.71 | – |
| 9.755 | 1,2-boxylic acid, bis 2-methylpropyl) ester | ester | $C_{16}H_{22}O_4$ | 278.34 | 0.76 | Cell proliferation inhibitor, antidiabetic [49] |
| 10.292 | 7,9-Di-tert-butyl-1-oxaspiro(4,5)deca-6,9-diene-2,8-dione | Lactone | | | 1.30 | – |
| 10.431 | Hexadecanoic acid, methyl ester | Fatty acid ester | $C_{18}H_{36}O_2$ | 284 | 0.50 | Anti-bacterial, antioxidant, antihyperlipidemic, 5- α reductase inhibitor [50] |
| 11.217 | Hexadecanoic acid, ethyl ester | Fatty acid ester | $C_{18}H_{36}O_2$ | 284.5 | 2.91 | Antioxidant, anti- inflammatory [51] |
| 11.555 | n-Hexadecanoic acid | Saturated fatty acid | $C_{16}H_{32}O_2$ | 256.42 | 3.07 | Antioxidant, anti-inflammatory [51] |
| 12.208 | Nonanedioic acid, dibutyl ester | Saturated fatty acid | $C_{17}H_{32}O_4$ | 300.43 | 0.27 | Anti-bacterial [52] |
| 12.594 | Cetene | Alkene | $C_{16}H_{32}$ | 224.42 | 3.96 | Anti-microbial, antioxidant [53] |
| 12.915 | Phytol | Diterpenoid | $C_{20}H_{40}O$ | 296.5 | 0.40 | Antioxidant, anti-inflammatory [22] |
| 13.489 | Hexadecanoic acid, 2-methylpropyl ester | Ester | $C_{20}H_{40}O_2$ | 312.5 | 0.29 | – |
| 13.755 | Linoleic acid ethyl ester | Fatty acid ethyl ester | $C_{20}H_{36}O_2$ | 308.5 | 0.06 | Antioxidant, anti-inflammatory [51] |
| 13.900 | (Z,Z,Z)-6,9,15-Octadecatrienoic acid methyl ester | Fatty acid ester | $C_{19}H_{32}O_2$ | 292.5 | 0.14 | – |
| 14.202 | Butanoic acid, 2-hydroxy-2-(1-methoxyethyl)-3-methyl-, (2,3,5,7a-tetrahydro-1-hydroxy-1H-pyrrolizin-7-yl)methyl ester, [1S-[1. α,7(2 R*,3S*),7a. α]]- | Ester | $C_{16}H_{27}NO_5$ | 313.39 | 0.44 | – |
| 14.552 | Hexadecanoic acid, butyl ester | Ester | $C_{20}H_{40}O_2$ | 312.5 | 8.71 | Antioxidant [54] |
| 14.945 | Heptadecyl acetate | Carboxylic ester | $C_{19}H_{38}O_2$ | 298.5 | 0.36 | – |
| 16.649 | Heptadecanoic acid, butyl ester | Ester | $C_{21}H_{42}O_2$ | 326.6 | 0.39 | – |
| 17.821 | Butyl 9,12-octadecadienoate | Ester | $C_{22}H_{40}O_2$ | 336.6 | 0.37 | – |
| 17.948 | n-Propyl 11-octadecenoate | Ester | $C_{21}H_{40}O_2$ | 324.5 | 0.78 | Antioxidant, anti-cancer [54] |
| 18.032 | 1,2-Benzisothiazole, 3-(hexahydro-1H-azepin-1-yl)-, 1,1-dioxide | Benzothiazole heterocyclic nucleus | $C_{13}H_{16}N_2O_2S$ | 264.35 | 0.14 | Anti-diabetic, anti-inflammatory [25] |
| 18.383 | Octadecanoic acid, butyl ester | Fatty acid ester | $C_{22}H_{44}O_2$ | 340.6 | 1.06 | Anti-microbial, anti-inflammatory, hypocholestrolemic [55] |
| 18.908 | 1-Hexacosanol | Phytosterol | $C_{26}H_{54}O$ | 382.7 | 0.08 | Larvicidal [56] |
| 19.198 | 6. α-Pentyl-4-oxa-5.beta.-androstane-3,17-dione | ketone | $C_{23}H_{36}O_3$ | 360.5 | 0.06 | – |
| 19.791 | Octacosyl acetate | Ester | $C_{30}H_{60}O_2$ | 452.8 | 0.25 | – |
| 20.437 | Cis-10-Nonadecenoic acid | Carboxylic acid | $C_{19}H_{36}O_2$ | 296.5 | 0.06 | Anti-cancer [57] |

*(Continued)*

Table 3. (Continued)

| RT (min) | Compound name | Compound class | Molecular formula | Molecular mass | Area | Biological activity |
|---|---|---|---|---|---|---|
| 20.564 | Cycloeicosane | Phenol | $C_{20}H_{40}$ | 280.5 | 0.14 | – |
| 20.818 | Arachidic acid | Long chain fatty acid | $C_{20}H_{40}O_2$ | 312.5 | 0.17 | – |
| 21.984 | Cyclohexane, (2-ethyl-1-methyl-1-butenyl)- | Cycloalkane | $C_{13}H_{24}$ | 180.33 | 0.14 | – |
| 22.667 | Docosanoic acid, butyl ester | Ester | $C_{26}H_{52}O_2$ | 396.7 | 0.41 | – |
| 23.041 | α-Tocospiro B | Sesterterpenoid | $C_{29}H_{50}O_4$ | 462.7 | 0.78 | Anti tuberculosis [58] |
| 24.256 | Tetracosanoic acid, butyl ester | Ester | $C_{28}H_{56}O_2$ | 424.7 | 0.65 | – |
| 24.395 | 1-Triacontanol | Fatty alcohol | $C_{30}H_{62}O$ | 438.8 | 0.10 | – |
| 24.395 | Tetracosanal | Fatty aldehyde | $C_{24}H_{48}O$ | 352.6 | 0.10 | – |
| 24.781 | Ergost-7-en-3-ol, (3.ß.)- | Ergosteroid | $C_{28}H_{48}O$ | 400.7 | 0.18 | Anticancer [59] |
| 25.186 | Vitamin E | Tocopherol | $C_{29}H_{50}O_2$ | 430.7 | 0.42 | Antioxidant [60] |
| 25.851 | Ergost-5-en-3-ol, (3. ß.)- | Ergosteroid | $C_{28}H_{48}O$ | 400.7 | 0.97 | – |
| 26.038 | Stigmasterol | Phytosterol | $C_{29}H_{48}O$ | 412.7 | 1.43 | Antifungal, antimutagenic, antidiabetic, antitumor, antiosteoarthritic and anti-inflammatory [61] |
| 26.570 | γ – Sitosterol | Phytosterol | $C_{29}H_{50}O$ | 414.7 | 23.23 | Antidiabetic, Anticancer [62] |
| 26.842 | ß-Sitosterol | Phytosterol | $C_{29}H_{50}O$ | 414.7 | 0.23 | Anti-microbia [63] |
| 27.035 | Ethyl triacontanate | Hydrocarbon | $C_{32}H_{64}O_2$ | 480.8 | 3.58 | – |
| 27.234 | Stigmasta-3,5-diene | Phytosterol | $C_{29}H_{48}$ | 396.7 | 1.44 | Antioxidant [63] |
| 27.307 | Methyl (25RS)-3. ß -hydroxy-5-cholesten-26-oate | Hydrocarbon | $C_{28}H_{46}O_3$ | 430.7 | 0.24 | – |
| 28.322 | Benzenepropanoic acid, 3,5-bis-dimethylethyl)-4-hydroxy-, octadecyl ester | Ester | $C_{35}H_{62}$ | 530.86 | 9.58 | – |

glucose level. Moreover, it increases the antioxidant capacity of the cells and significantly improves hyperlipidemia [30]. The inclusion of Myricetin and Quercetin in ACEA may contribute to its strong hypoglycemic properties. The phenols are another class of physiologically active chemicals. They achieve their hypoglycemic impact by disrupting the metabolic pathways involved in carbohydrate processing, specifically gluconeogenesis, glycogenesis, and glycolysis. The HPLC analysis of the ACNB extract indicated the presence of sinapic acid, which enhances the activity of the enzyme glucokinase, leading to a rise in insulin levels. It mitigates the inflammation and oxidative stress linked to diabetes by promoting the regeneration of the β-cells [31].

## Oral glucose tolerance test

The assessment of insulin secretion and insulin resistance can be conducted by the utilization of the commonly employed oral glucose tolerance test (OGTT). The findings from the oral glucose tolerance test (OGTT), as shown in Table 5, indicate that blood glucose levels peaked 30 minutes after glucose administration, followed by an enhancement in glucose tolerance after 60 minutes ($p < 0.05$). ACEA and ACNB demonstrated significant outcomes at doses of 500 and 1000 mg/kg in comparison to the disease control group. ACEA and ACNB had glucose-lowering efficacy like that of the conventional medication glibenclamide at 60 and 120 minutes. Extensive clinical trials and epidemiologic investigations have conclusively demonstrated that hyperglycemia is the primary cause of diabetes complications. The condition can be

**Table 4. Phytochemical components identified in ACNB by GC-MS analysis.**

| RT (min) | Compound name | Compound class | Molecular formula | Molecular mass | Area | Biological activity |
|---|---|---|---|---|---|---|
| 3.187 | L-Alanine, N-methyl- | Amino acid | $C_4H_9NO_2$ | 103.12 | 16.15 | – |
| 5.634 | 1,4-Benzenedicarboxylic acid, dimethyl ester | Diester | $C_6H_4(COOCH_3)2$ | 194.18 | 0.33 | Antimicrobial, antioxidant [49] |
| 8.250 | Isomethadone | Di aryl methane | $C_{21}H_{27}NO$ | 309.4 | 0.14 | Analgesic [64] |
| 8.655 | 4-Amino-1-hexanol | Amino alcohol | $C_6H_{15}NO$ | 117.19 | 0.92 | – |
| 9.078 | Guaifenesin | Methoxy benzene | $C_{10}H_{14}O_4$ | 198.22 | 3.49 | Antidiabetic [26] |
| 9.441 | Bicyclo[3.1.1]heptane, 2,6,6-trimethyl-, [1R-(1.α,2. ß,5. ß)]- | terpene | $C_{10}H_{18}$ | 138.25 | 1.52 | Antimicrobial [65] |
| 9.507 | Stearic Acid | Fatty acid | $CH_3(CH_2)_{16}COOH$ | 284.5 | 1.07 | Antimicrobial, antidepressant [66] |
| 10.389 | Hexadecanoic acid | Fatty acid | $C_{16}H_{32}O_2$ | 256.42 | 3.17 | Antioxidant, anti-inflammatory [67] |
| 12.371 | 1-Hexacosanol | Fatty alcohol | $C_{26}H_{54}O$ | 382.7 | 0.25 | Larvicidal [68] |
| 12.371 | 1-Octadecene | Hydrocarbon | $C_{18}H_{36}$ | 252.5 | 0.25 | Anti-bacterial, antioxidant, anti-cancer [69] |
| 12.371 | Heptacosyl acetate | Carboxylic ester | $C_{29}H_{58}O_2$ | 438.8 | 0.25 | – |
| 12.456 | 9,11-Octadecadienoic acid, methylester, (E,E)- | Ester | $C_{19}H_{34}O_2$ | 294.5 | 0.45 | Anti-cancer, anti-inflammatory, hypocholestrolemic, hepato-protective [67] |
| 12.576 | 10-Octadecenoic acid, methyl ester | Ester | $C_{19}H_{36}O_2$ | 296.5 | 1.02 | Antibacterial, antifungal, antioxidant, decrease blood cholesterol [67] |
| 12.576 | 13-Octadecenoic acid, methyl ester | Fatty acid Ester | $C_{19}H_{36}O_2$ | 296.5 | 1.02 | Antimicrobial [67] |
| 12.576 | 9-Octadecenoic acid, methyl ester, (E)- | Fatty acid Ester | $C_{19}H_{36}O_2$ | 296.5 | 1.02 | Antioxidant, anti-cancer [67] |
| 12.740 | Phytol | diterpenoid | $C_{20}H_{40}O$ | 296.5 | 1.65 | Antioxidant, anti-inflammatory [22] |
| 13.042 | Heptadecanoic acid, 16-methyl-ester | Fatty acid ester | $C_{18}H_{36}O_2$ | 284.5 | 0.42 | – |
| 13.042 | Pentadecanoic acid, methyl ester | Fatty acid ester | $C_{16}H_{32}O_2$ | 256.42 | 0.42 | Antimicrobial, Antioxidant antifungal [67] |
| 14.196 | Hexadecanoic acid, butyl ester | Fatty acid ester | $C_{20}H_{40}O_2$ | 312.5 | 2.16 | Antimicrobial, antioxidant [67] |
| 14.196 | Hexadecanoic acid, 1,1-dimethylethyl ester | Fatty acid ester | $C_{10}H_{20}O2$ | 172.2 | 2.16 | – |
| 16.782 | 1-Heptadecene | Alkane | $C_{17}H_{36}$ | 240.5 | 1.15 | Antioxidant, antimicrobial, anticancer [69] |
| 17.501 | 2(3H)-Furanone, dihydro-4,4-dimethyl- | β-unsaturated Lactone | $C_6H_{10}O_2$ | 114.4 | 0.27 | – |
| 17.766 | 2-Methyl-Z,Z-3,13-octadecadienol | Hydrocarbon | $C_{19}H_{36}O$ | 280.5 | 0.64 | Pheromone, pesticide, insecticide, herbicide [69] |
| 18.226 | Octadecanoic acid, butyl ester | Fatty acid ester | $C_{22}H_{44}O_2$ | 340.6 | 0.20 | |
| 19.132 | 2(5H)-Furanone, 3-chloro-5-((dimethylamino)methyl-4,5-dimethyl- | Lactone | $C_9H_{14}ClNO_2$ | 206.66 | 0.13 | – |
| 20.383 | 1,2-Benzisothiazole, 3-(hexahydro-1H-azepin-1-yl)-, 1,1-dioxide | Benzothiazole | $C_{13}H_{16}N_2O_2S$ | 264.35 | 0.25 | Anti-diabetic, anti-inflammatory [25] |
| 20.383 | 1-Tridecene | Hydrocarbon | $C_{13}H_{26}$ | 182.35 | 0.25 | Antibacterial [70] |
| 20.951 | 7 Oxabicyclo[4.1.0]heptane, 1,5-dimethyl- | Oxabicyclic compound | $C_8H_{14}O$ | 126.20 | 0.16 | – |
| 21.283 | 4,6-Decadienal, 8-ethyl-10-[4-hydroxy-8-(2-hydroxypropyl)-3,9-dimethyl-1,7-dioxaspiro[5.5]undec-2-yl]- 2-methyl- | Hydrocarbon | $C_{27}H_{46}O_5$ | 450.7 | 0.17 | – |

*(Continued)*

| RT (min) | Compound name | Compound class | Molecular formula | Molecular mass | Area | Biological activity |
|---|---|---|---|---|---|---|
| 21.283 | Erucic acid | Fatty acid | $C_{22}H_{42}O_2$ | 338.6 | 0.17 | Antioxidant, anti-inflammatory, hypolipidemic [71] |
| 22.081 | Octadecanoic acid, 12-hexyl-, methyl ester | Methyl ester | $C_{25}H_{50}O_2$ | 382.7 | 0.24 | – |
| 23.204 | Methyl 2-hydroxy-tetracosanoate | Methyl ester | $C_{25}H_{50}O_3$ | 398.7 | 0.73 | – |
| 24.316 | Hexadecanoic acid, octadecyl ester | Methyl ester | $C_{34}H_{68}O_2$ | 508.9 | 0.20 | – |
| 25.108 | dl-α-Tocopherol | Phenol | $C_{29}H_{50}O_2$ | 430.7 | 0.13 | Antioxidant [60] |
| 26.340 | β -Sitosterol | Phytosterol | $C_{29}H_{50}O$ | 414.7 | 4.77 | Antimicrobial [63] |
| 26.340 | γ-Sitosterol | Phytosterol | $C_{29}H_{50}O$ | 414.7 | 4.77 | Antidiabetic, Anticancer [62] |
| 26.527 | 9,19-Cycloergost-24(28)-en-3-ol, 4,14-dimethyl-, acetate, (3. ß,4. α,5. α)- | Acetate | $C_{32}H_{52}O_2$ | 468.8 | 0.53 | |
| 26.914 | Stigmasta-3,5-dien-7-one | Stigmastan type steroid | $C_{29}H_{46}O$ | 410.7 | 1.20 | Synthetic progesterone [72] |
| 27.222 | Octasiloxane, 1,1,3,3,5,5,7,7,9,9, 11,11,13,13,15,15-hexadecamethyl- | Silicon ether | $C_{16}H_{48}O_7Si_8$ | 577.2 | 0.30 | Antimicrobial [73] |
| 27.428 | Acetic acid, 17-(1,5-dimethyl-hexyl)-4,4,10,13,14-pentamethyl-2,3,4, 5,8,10,12,13,14,15,16,17-dodecahydro-1H-cyclopenta[a]phenanthren-3-ol (ester) | Ester | $C_{32}H_{52}O_2$ | 468.8 | 1.26 | – |
| 27.778 | Pyridine-3-carboxamide, oxime, N-(2-trifluoromethylphenyl)- | oxime | $C_{13}H_{10}F_3N_3O$ | 281.23 | 0.98 | Antioxidant, antibacterial, antifungal [74] |
| 27.899 | Methyl 3.beta.-hydroxyolean-18-en-28-oate | Ester | $C_{31}H_{50}O_3$ | 470.7 | 3.11 | – |
| 20.226 | Flexibilide | Terpene | $C_{20}H_{30}O_4$ | 334.4 | 0.28 | Anti-inflammatory, cytotoxic, antimicrobial [75] |
| 28.117 | Antra-9,10-quinone, 1-(3-hydrohy-3-phenyl-1-triazenyl)- | Amine | $C_{20}H_{13}N_3O_3$ | 343.3 | 0.14 | – |

prevented or reversed with continuous lowering or regulation of blood glucose levels. This served as the foundation for choosing a diabetes model that was loaded with glucose to evaluate the potential of plant extracts as antidiabetic agents. The main biochemical marker utilized in clinical and experimental settings to diagnose and track the progression of diabetes mellitus is the concentration of glucose in the blood or serum/plasma [32]. The findings of our investigation revealed that the blood glucose levels of untreated diabetic rats were elevated during the oral glucose tolerance test in comparison to untreated healthy rats. The results align with the findings of other writers who conducted experiments on rats with diabetes induced by alloxan.

**Anti-hyperglycemic effect of active extracts of *A. crassifolia* in alloxan induced hyperglycemic rats**

Fig 5 demonstrates the anti-hyperglycemic benefits of *A.crassifolia* extracts. The animals identified a significant increase in blood glucose levels, on 3rd day after the administration of alloxan at a dosage of 150 mg/kg. The groups treated with ACEA exhibited a decrease in glucose levels that was dependent on the dosage administered, as observed throughout the duration of the trial. No significant impact on hyperglycemia was observed at doses of 250 mg/kg/day of ACEA. Nevertheless, administering a dosage of 500 and 1000 mg/kg/day of ACEA resulted in a considerable reduction in blood glucose levels starting from the 9th day of the trial, and this decline continued steadily until the 28th day. ACNB, administered at doses of 250, 500, and 1000 mg/kg/day, eliminated the rise in blood glucose levels induced by alloxan. This effect was observed starting from day 9, with the maximum impact occurring on day 28. Glibenclamide, taken as standard drug,

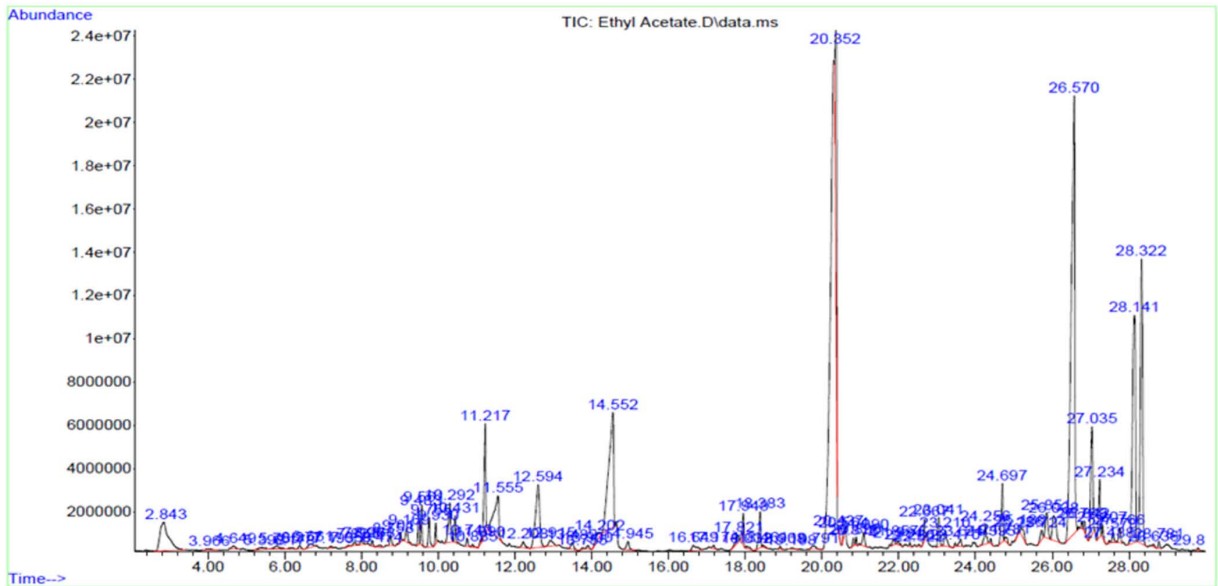

**Fig 1. GC-MS spectra of ACEA.**

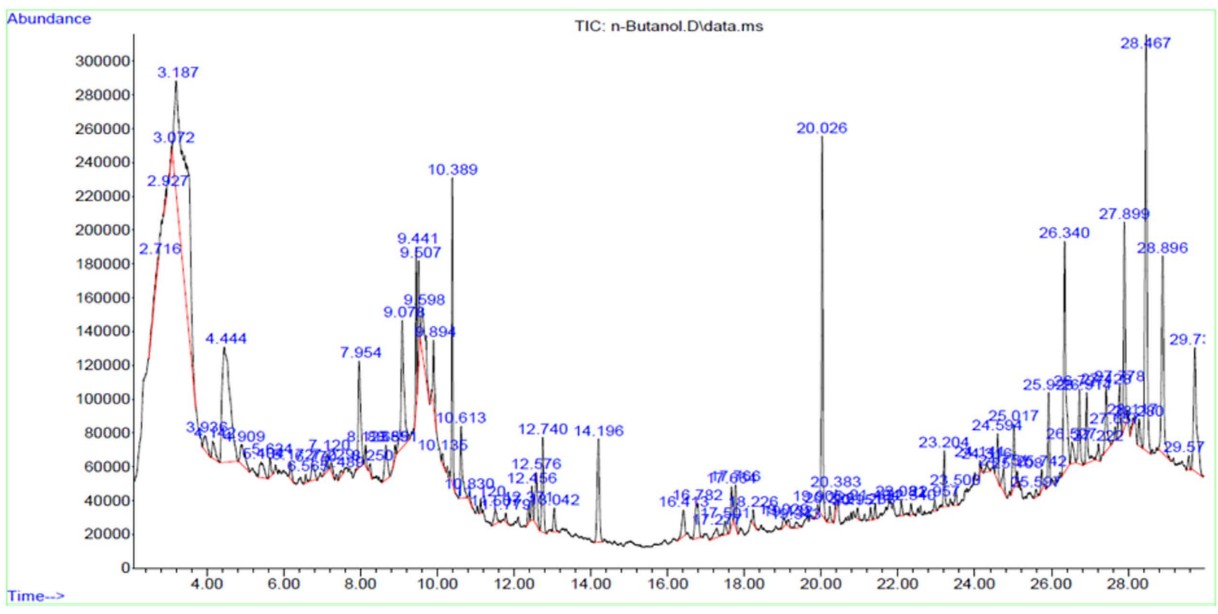

**Fig 2. GC-MS spectra of ACNB.**

effectively reversed high blood sugar levels, just like ACEA and ACNB. ACEA and ACNB demonstrated a dose-dependent hypoglycemic effect. This may correlate with the quantity of hypoglycemic phytoconstituents present, as demonstrated by GC-MS and HPLC analyses. Plant extracts demonstrate antidiabetic effects via many pathways, including modulation of glucose absorption, insulin secretion, and oxidative stress, contingent upon the type and concentration of bioactive

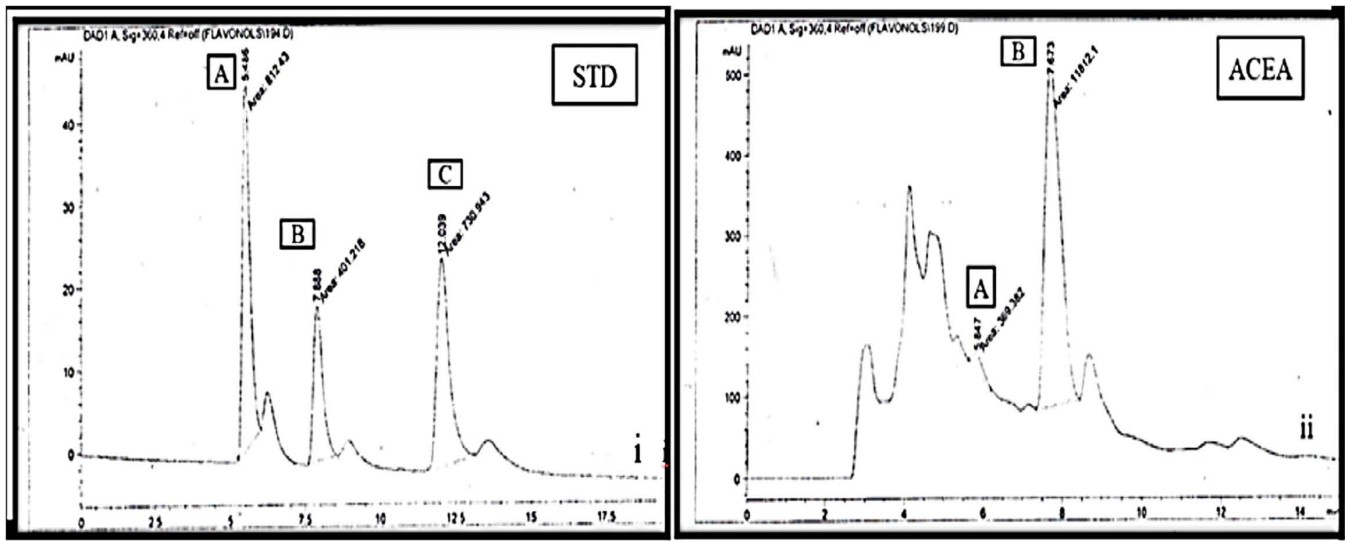

**Fig 3. HPLC chromatograms for the detection of flavonoids in ethyl acetate extract of *A. crassifolia* (i = standard, ii = ethyl acetate extract).**
ACEA: *A. crassifolia*, STD: standard, i: Chromatogram for standards for Myricetin(A), Quercetin(B) and Kaempferol(C), ii: Chromatogram for ACEA showing peaks for Myricetin (A) and Quercetin (B) compared with standard (i).

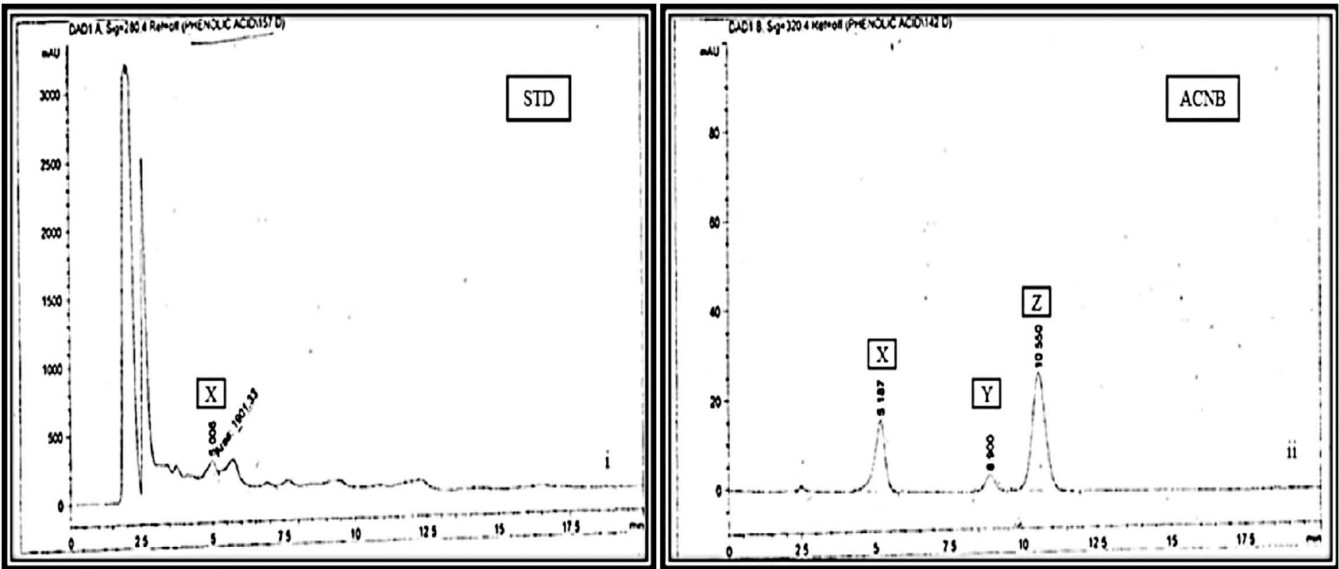

**Fig 4. HPLC chromatograms for the detection of phenols in n-butanol extract of *Atriplex crassifolia*.** ACNB: *A.crassifolia* n-butanol extract, STD: Standard, i.Chromatogram for phenol standards, Sinapic acid(X), Caffeic acid(Y), Gallic acid(Z), ii. Chromatogram for ACNB showing peaks for Sinapic acid (X) Compared with standard (i).

chemicals they contain [33]. This may influence the varied hypoglycemic response of ACEA and ACNB at varying dosages.The reduction in blood glucose levels can be attributed to the plant's ability to enhance the secretion of insulin from undamaged β cells. ACEA and ACNB may induce a similar increase of insulin secretion as glibenclamide. Reports indicate

**Table 5. Oral glucose tolerance test of different extracts of *A. crassifolia* in Alloxan induced diabetic rats.**

| Groups | Dose (mg/kg) | Blood Glucose level(mg/dl) | | | | |
|---|---|---|---|---|---|---|
| | | −30 minutes | Base line | 30 minutes | 60 minutes | 120 minutes |
| Normal control | D/W | $100.6\pm 0.55^{***}$ | $92.20\pm 1.00^{***}$ | $96.95\pm 0.15^{***}$ | $92.0\pm 0.80^{***}$ | $94.0\pm 0.40^{***}$ |
| Diabetic control | D/W | $264.75\pm 0.25$ | $264.4\pm 2.15$ | $263.7\pm 0.9$ | $231.8\pm 0.35$ | $261.5\pm 0.75$ |
| Standard control | 10 | $269.3\pm 1.14^{ns}$ | $209.9\pm 1.85^{***}$ | $204.8\pm 0.35^{***}$ | $116\pm 0.82^{***}$ | $146.7\pm 1.10^{***}$ |
| ACNH | 250 | $258.6\pm 0.42^{ns}$ | $261.1\pm 0.35^{ns}$ | $264.9\pm 0.42^{ns}$ | $244.3\pm 2.25^{ns}$ | $268.4\pm 0.20^{ns}$ |
| | 500 | $258.5\pm 3.05^{ns}$ | $269.7\pm 1.65^{ns}$ | $274.6\pm 0.40^{**}$ | $275.6\pm 0.55^{ns}$ | $253.5\pm 1.50^{ns}$ |
| | 1000 | $270.6\pm 0.40^{ns}$ | $264.5\pm 0.55^{ns}$ | $266.8\pm 0.45^{ns}$ | $253.6\pm 1.53^{**}$ | $253.6\pm 4.75^{ns}$ |
| ACDCM | 250 | $255.7\pm 0.5^{**}$ | $276.6\pm 0.36^{**}$ | $258.1\pm 0.58^{ns}$ | $242.9\pm 0.87^{ns}$ | $257.2\pm 0.39^{ns}$ |
| | 500 | $258\pm 0.59^{**}$ | $254.6\pm 1.0^{**}$ | $261.5\pm 1.26^{ns}$ | $240.5\pm 0.71^{ns}$ | $264.9\pm 0.43^{ns}$ |
| | 1000 | $257.9\pm 0.83^{**}$ | $250.9\pm 0.71^{**}$ | $258\pm 0.05^{ns}$ | $221.5\pm 3.0^{ns}$ | $229\pm 3.0^{**}$ |
| ACEA | 250 | $266.5\pm 0.55^{ns}$ | $259.8\pm 1.55^{ns}$ | $258.0\pm 0.92^{**}$ | $238.0\pm 1.86^{***}$ | $245.8\pm 0.95^{**}$ |
| | 500 | $259.2\pm 0.60^{ns}$ | $236.7\pm 0.52^{**}$ | $244.5\pm 1.30^{**}$ | $153.9\pm 1.32^{***}$ | $145\pm 0.40^{***}$ |
| | 1000 | $261.6\pm 10^{ns}$ | $244.4\pm 1.21^{*}$ | $243.2\pm 0.80^{**}$ | $153.3\pm 2.52^{***}$ | $136.8\pm 4.52^{***}$ |
| ACNB | 250 | $262.2\pm 0.71^{ns}$ | $241.9\pm 0.36^{**}$ | $262.5\pm 0.71^{ns}$ | $185.7\pm 9.96^{***}$ | $162.1\pm 3.25^{***}$ |
| | 500 | $263.0\pm 0.51^{ns}$ | $256.9\pm 0.32^{*}$ | $241.4\pm 1.18^{***}$ | $165.0\pm 0.51^{***}$ | $143.0\pm 0.32^{***}$ |
| | 1000 | $266.4\pm 0.61^{ns}$ | $255.9\pm 0.72^{*}$ | $250.3\pm 0.71^{***}$ | $121.6\pm 0.54^{***}$ | $112.2\pm 0.38^{***}$ |

Results are shown as mean±SEM. The results are compared with diabetic control.

ACNH: *A. crassifolia* n hexane extract, ACDCM: *A. crassifolia* Dichloromethane extract, ACEA: *A. crassifolia* Ethyl acetate extract, ACNB: *A. crassifolia* n-butanol extract, *=<0.01, **=<0.001, ***=<0.0001 respectively by one way ANOVA followed by Dunnet's multiple comparison test. (n = 5).

that glibenclamide stimulates insulin production by blocking ATP-sensitive potassium channels in the pancreatic β cells [34]. Extracts of *A. crassifolia* may possess the capacity to promote the regeneration of β cells damaged by alloxan administration and improve glucose absorption and utilization by tissues. The medicinal properties of most plants are ascribed to their strong antioxidant potential. Free radicals are crucial in the pathophysiology of numerous illnesses. Phytoconstituents, specifically phenols and flavonoids, function as optimal exogenous antioxidants. They not only inhibit reactive oxygen species (ROS) but also enhance the body's intrinsic defense mechanisms against free radicals. Plants containing phenols and flavonoids have a broader spectrum of biological activities compared to traditional hydrogen-donating substances [35]. The antioxidant properties of *A. crassifolia* may account for its notable antidiabetic effects in our study. A robust link exists between oxidative stress in diabetes and a reduction in antioxidant levels. Oxidative stress induces increased generation of reactive oxygen species (ROS), concurrently diminishing the antioxidative defense mechanisms in diabetic animals [34]. Plants rich in phenols, terpenes, and flavonoids have been shown to mitigate oxidative stress,

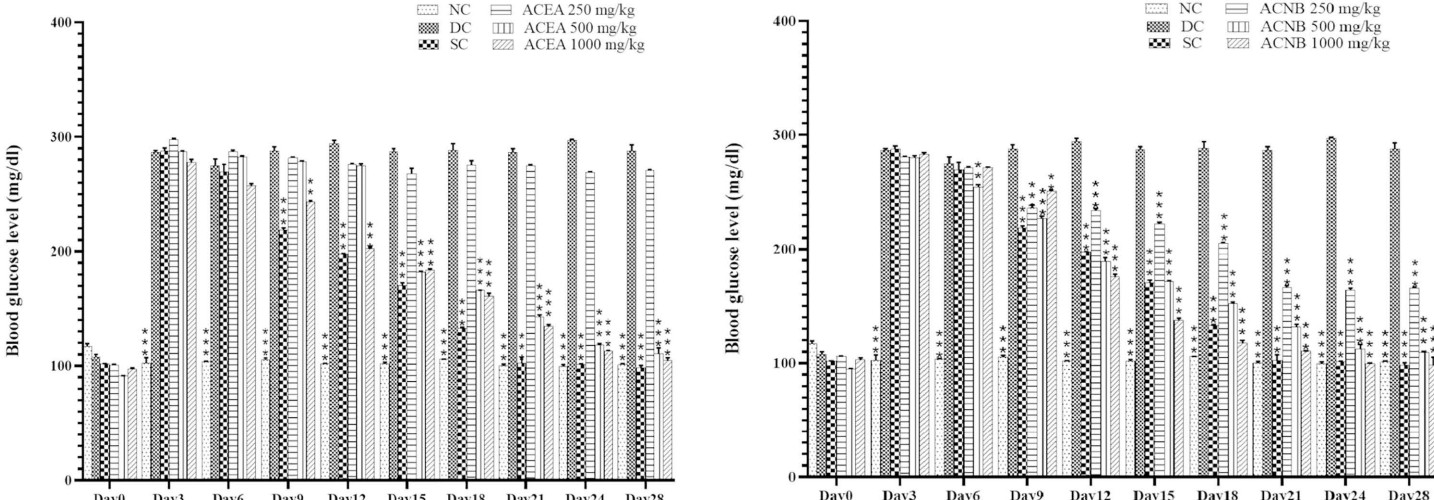

**Fig 5. Effects of active extracts of *Atriplex crassifolia* on blood glucose levels of alloxan induced diabetic rats.** Results are shown as mean ± SEM, ACEA: *A. crassifolia* Ethyl acetate extract, ACNB: *A. crassifolia* n-butanol extract, *=<0.01, **=<0.001, ***=<0.0001 respectively by one way ANOVA followed by Dunnet's multiple comparison test.

hence avoiding or decelerating the advancement of diseases such as diabetes. Consequently, plants possessing elevated antioxidant capacity significantly enhance human health and physical condition. This results from their improved therapeutic efficacy attributed to the presence of antioxidant phytoconstituents [35]. A crucial aspect of general well-being is the improvement of metabolic health. Meta-analysis of randomized controlled trials has substantiated clinical evidence that antioxidant supplementation in obese persons significantly lowers fasting blood glucose levels, enhances insulin resistance, and diminishes lipid peroxidation [36]. Flavonoids possess a stable structure that enables them to function as free radical scavengers and inhibit oxidases. They regulate oxidative stress levels in the body below a crucial threshold, hence playing an essential role in the management of numerous neurodegenerative, inflammatory, and metabolic illnesses [37]. The HPLC analysis of several extracts of *A. crassifolia* revealed the presence of phenols and flavonoids. The GC-MS study additionally revealed the existence of antioxidant compounds, which may account for the plant's biological activity.

### Effect of active extracts of *A.crassifolia* on weight of hyperglycemic rats

Diabetes mellitus can cause weight loss as a physical symptom. The study animals' weights were measured weekly until the 28th day of the trial to assess the impact of ACEA and ACNB extracts (Table 6). At first, the rats' weights declined fast because of the advancement of diabetes. The control animals exhibited a rise in body weight throughout the study period, but the diabetes control group saw a significant decrease in weight. The weight reduction associated with diabetes can be linked to the depletion of structural proteins [38].

After the 21st day of treatment, a dose-dependent positive shift in weight increase was observed. The results were highly significant in ACEA and ACNB treated groups (P < 0.05) at 1000 mg/kg, when compared with the diseased control group. The improvements in body weight after the administration of plant extracts support that these extracts possess both pancreatic and extra-pancreatic mechanisms of action to reduce the blood glucose levels.

### Assessment of metabolic biomarkers in various experimental groups

An elevation in total cholesterol (TC), triglycerides (TG), low-density lipoprotein (LDL), very low-density lipoprotein (vLDL), and a reduction in high-density lipoprotein (HDL) levels is indicative of lipid metabolic problems associated with diabetes

**Table 6. Weight variation in various groups treated with active _A. crassifolia_ extracts.**

| Groups | Dose (mg/kg) | Weight | | | | |
|---|---|---|---|---|---|---|
| | | 0 | 7 | 14 | 21 | 28 |
| Normal control | Vehicle | 251± 1.02** | 271.5± 6.54*** | 266.5± 1.59*** | 282.5± 3.5*** | 300± 2.19*** |
| Diabetic control | Vehicle | 231± 1.05 | 209± 1.23 | 170± 2.00 | 166± 2.0 | 146± 1.54 |
| Standard control | 10 | 224± 1.23ns | 196± 1.51* | 183.5± 1.5ns | 174± 2.0ns | 173.5± 1.5*** |
| ACEA | 250 | 227.5± 2.52ns | 210.5± 2.51ns | 184± 2.20ns | 170± 2.0ns | 172.5± 1.59ns |
| | 500 | 221± 1.02ns | 195.5± 2.51ns | 183.9± 1.59ns | 173.8± 1.5ns | 179.6± 1.02** |
| | 1000 | 220± 5.0ns | 200± 2.23ns | 182.5± 2.56ns | 185.5± 2.02*** | 186.2± 1.15*** |
| ACNB | 250 | 226.5± 2.45ns | 210.5± 0.51ns | 186± 3.20* | 175.8± 0.05ns | 170.7± 0.45ns |
| | 500 | 229± 1.10ns | 212.5± 2.58ns | 187± 2.23* | 176.9± 3.5ns | 174.4± 2.1ns |
| | 1000 | 230± 1.25ns | 184.5± 1.59* | 175± 4.09ns | 181.7± 0.2** | 185.1± 1.35*** |

Results are shown as mean±SEM, ACEA: _A. crassifolia_ Ethyl acetate extract, ACNB: _A. crassifolia_ n-butanol extract, *=<0.01, **=<0.001, ***=<0.0001 respectively by one way ANOVA followed by Dunnet's multiple comparison test (n=5).

mellitus. Insulin significantly influences lipid metabolism by suppressing the principal enzyme responsible for cholesterol synthesis. This complication of diabetes arises from heightened mobilization of fatty acids from adipose tissue. Insulin inhibits the action of HMG-CoA, the essential enzyme responsible for cholesterol synthesis. Increased blood cholesterol levels arise from insulin insufficiency or insulin resistance [39]. The study found that the disease group had elevated blood cholesterol levels and an overall high lipid profile (Fig 6). Treatment of alloxan induced diabetic animals with different extracts of _A. crassifolia_ particularly ACEA and ACNB, exhibited contrasting effects compared to the diabetic control group (p<0.05), as indicated by a reduction in blood TC, TG, LDL, and vLDL, with an elevation in HDL levels. This was evidenced by a decrease in total cholesterol, triglycerides, low-density lipoprotein, and very low-density lipoprotein levels, alongside an elevation in high-density lipoprotein levels (S1 Table in S1 File). Glibenclamide produced similar results. The extracts of _A. crassifolia_ unequivocally exhibit hypolipidemic effects, offering alleviation. The potential explanation may be diminished cholesterol absorption in the intestines or a reduction in HMG-CoA reductase activity, resulting in decreased cholesterol production [39].

Urea and creatinine are considered as significant biomarkers to estimate renal dysfunction (S2 Table in S1 File). Diabetic complications affect kidneys thereby causing the elevation in plasma levels of urea and creatinine. Our results (Fig 7) displayed a rise in level of plasma urea and creatinine in diabetic control group. This gives an estimation that diabetes elicits renal dysfunction [40]. Treatment of diabetic groups with extracts of _A.crassifolia_ reverted the levels of urea and creatinine towards normal when compared to the mean values of diabetic group (p<0.05) shown in Fig 7. This validates the utility of _A.crassifolia_ in the management of complications accompanying diabetes.

Diabetes is linked to alterations in metabolic function, resulting in aberrant variations in the activity of blood enzymes involved in metabolism. Numerous researchers have documented an elevation in liver enzyme activity in hyperglycemic animal models. In diabetes, heightened ketogenesis and gluconeogenesis elevate amino acid activity, thus increasing transaminase levels [41]. Increased activity of liver enzymes like aspartate transaminase (AST), alanine aminotransferase (ALT) and alkaline phosphatase (ALP) in present study indicates liver dysfunction caused by hyperglycemia (S2 Table

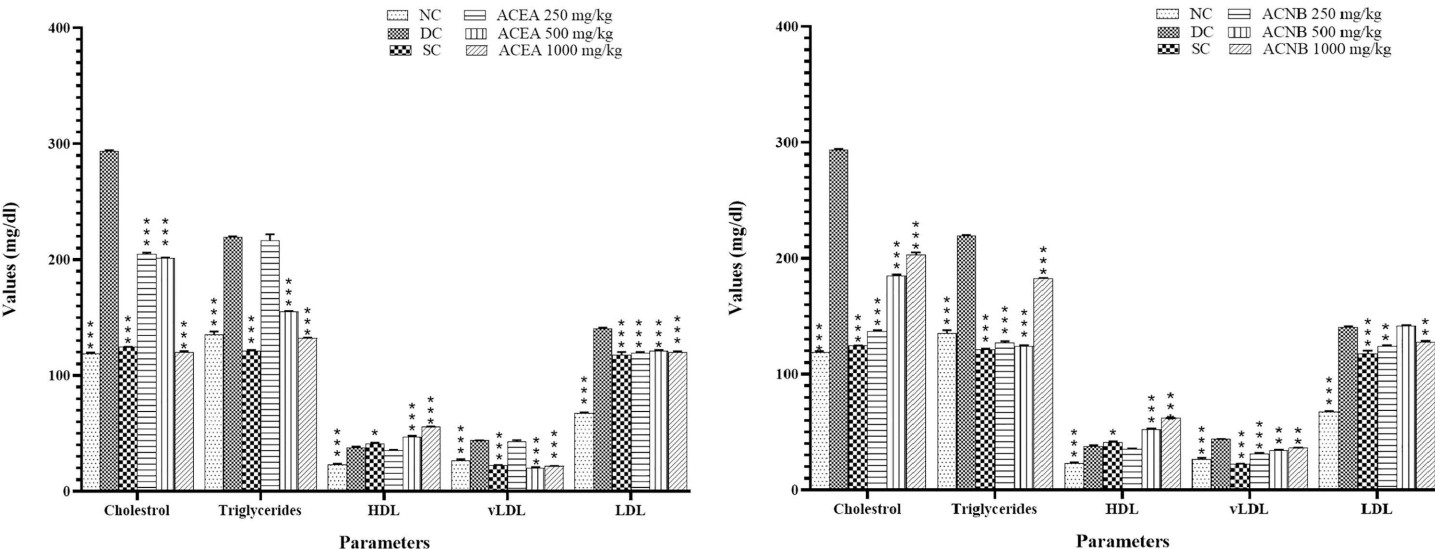

**Fig 6. Effects of active extracts of *Atriplex crassifolia* on lipid profiles of diabetic rats.** Results are shown as mean±SEM, ACEA: *A. crassifolia* Ethyl acetate extract, ACNB: *A. crassifolia* n-butanol extract, *=<0.01, **=<0.001, ***=<0.0001 respectively by one way ANOVA followed by Dunnet's multiple comparison test.

in S1 File). Diabetic control group displayed high plasma levels whereas ACEA and ACNB at all doses (250, 500 and 1000 mg/kg/day) depicted a fall in liver enzymes towards normal comparable to glibenclamide as presented in Fig 7. This further displays the efficacy of *A.crassifolia* extracts in ruling diabetes associated complications.

The histological study demonstrated that administration of ACEA at a dosage of 500 mg/kg effectively protected the pancreas from damage, as illustrated in Fig 8F. The acinar cells, characterized by a square shape, and the interlobular ducts, distinguished by a triangular shape, remain intact. Compared to the diabetic control group, the cellular architecture of the pancreatic islet, denoted by an arrow, is preserved, as illustrated in Fig 8B. Administering a dose of 500 mg/kg in ACNB maintained the cellular architecture. The islets, acinar cells, and interlobular ducts are intact, as illustrated in Fig 8H. Diabetes cause pancreatic impairment and diminishes islet cell functioning. The administration of alloxan induces chemical diabetes and results in oxidative damage to the islets of Langerhans [42]. The extracts successfully reduced cellular damage by potentially reducing oxidative stress.

## Conclusion

This study clearly demonstrates that ACEA and ACNB extracts of *A. crassifolia* exhibit glucose lowering effect and positively influence diabetes-induced hyperlipidemia and related complications. The biological benefits may be ascribed to the presence of bioactive substances, including phenols, flavonoids, terpenes, and phytosterols, as indicated by phytochemical analysis. Nonetheless, the study presents certain limitations as it fails to elucidate the methods and target receptors by which the plant compounds produce their effects. Advanced investigations, including molecular docking, pathway-specific elucidations, and receptor binding analysis, are necessary to discover and validate the target sites of active compounds. Furthermore, future research on *Atriplex crassifolia* should investigate the interacting mechanisms of its phytochemicals with various antidiabetic agents and the elucidation of hypoglycemia pathways. The current findings corroborate the hypoglycemic effects of *Atriplex crassifolia* extracts; however, dose-response experiments were not included in this research. Subsequent research may be conducted to determine dose-dependent effects to more accurately delineate the medicinal potential of the plant.

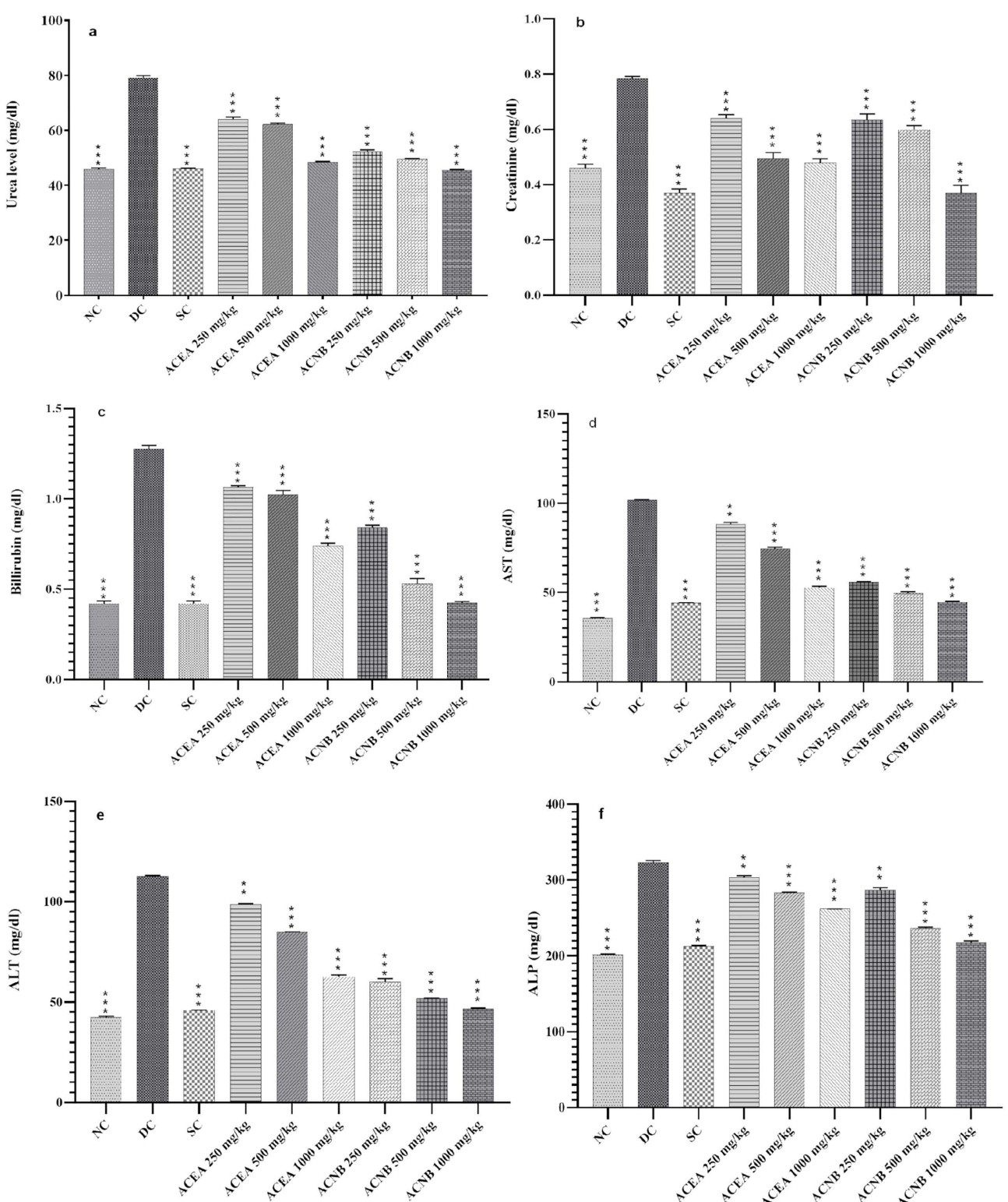

**Fig 7. Biochemical parameters of normal and hyperglycemic rats treated with active extracts of *A. crassifolia*.** Results are shown as mean±SEM, ACEA: *A. crassifolia* Ethyl acetate extract, ACNB: *A. crassifolia* n-butanol extract, *=<0.01, **=<0.001, ***=<0.0001 respectively by one way ANOVA followed by Dunnet's multiple comparison test. a: effect on urea, b: effect on creatinine, c: effect on bilirubin d: effect on aspartate transaminase e: effect on alanine aminotransferase, f: effect on alkaline phosphatase.

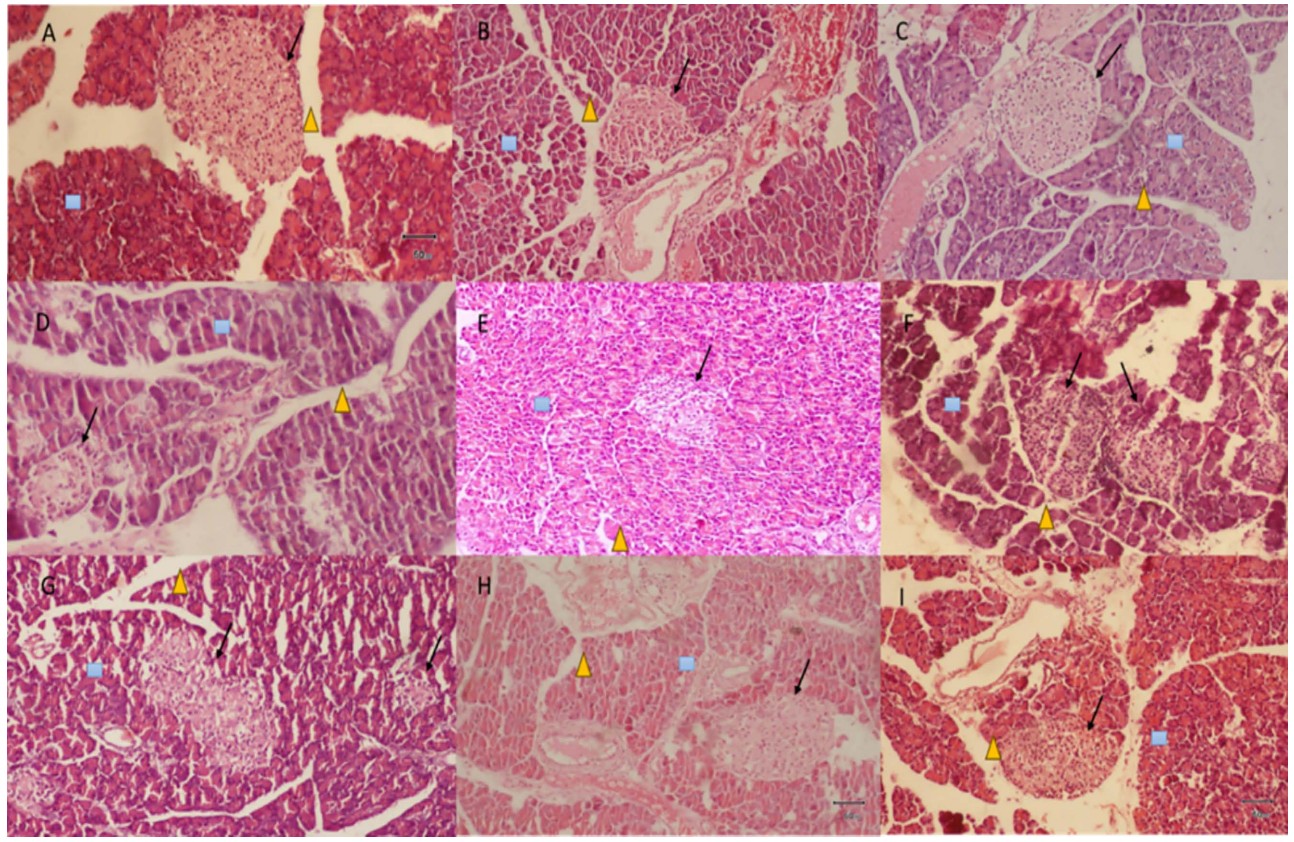

**Fig 8. Histopathological examination of pancreas excised from various experimental groups.** A: Normal control, B: Diabetic control, C: Standard control, D: ACEA 250 mg/kg, E: ACEA 500 mg/kg, F: ACEA 1000 mg/kg, g: ACNB 250 mg/kg, H: ACNB 500 mg/kg, I: ACNB 1000 mg/kg1000 mg/kg, g:

ACNB 250 mg/kg, H: ACNB 500 mg/kg, I: ACNB 1000 mg/kg 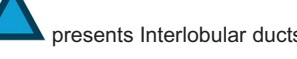 presents Interlobular ducts, ⬛ presents Acinar cells and Arrows represent Pancreatic islets of Langerhans.

## Supporting information

**S1 File. Hypoglycemic study complete data.**
(ZIP)

**S2 File. Supplemetary file of A. crassifolia.**
(DOCX)

## Author contributions

**Conceptualization:** Saiqa Ishtiaq.

**Data curation:** Sarah Rehman, Syeda Farheen Fatima.

**Formal analysis:** Sarah Rehman, Syeda Farheen Fatima.

**Investigation:** Sarah Rehman.

**Methodology:** Sairah Hafeez Kamran, Muhammad Khalil-ur-Rehman.

**Project administration:** Saiqa Ishtiaq, Sairah Hafeez Kamran.

Resources: Saiqa Ishtiaq, Muhammad Khalil-ur-Rehman.

Supervision: Saiqa Ishtiaq, Sairah Hafeez Kamran.

Visualization: Sairah Hafeez Kamran.

Writing – original draft: Sarah Rehman.

Writing – review & editing: Saiqa Ishtiaq, Sairah Hafeez Kamran, Muhammad Khalil-ur-Rehman, Numera Arshad, Saira Rehman.

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
