## [Decision Letter · Decision Letter 0]

18 Oct 2024

PONE-D-24-40010Investigation of phytochemical and biochemical attributes of hypoglycemic activity of Atriplex crassifolia (C.A. Mey) extracts in alloxan diabetic animal modelPLOS ONE

Dear Dr. Hafeez Kamran,

Thank you for submitting your manuscript to PLOS ONE. After careful consideration, we feel that it has merit but does not fully meet PLOS ONE’s publication criteria as it currently stands. Therefore, we invite you to submit a revised version of the manuscript that addresses the points raised during the review process.

We look forward to receiving your revised manuscript.

Kind regards,

Rajesh Kumar Singh, Ph.D.

Academic Editor

PLOS ONE

Additional Editor Comments:

The manuscript entitled “Investigation of phytochemical and biochemical attributes of hypoglycemic activity of Atriplex crassifolia (C.A. Mey) extracts in alloxan diabetic animal model” is interesting although poorly written and presentation is very poor. Many data are improperly analysed and presented, such as the antioxidant activity of extract was evaluated with highest dose 1000 µg/ml; how the IC50 values of n-hexane extract, and dichloromethane extract of A. crassifolia was evaluated, even higher than the highest concentration of experimentation? The figures in the manuscript are very poor quality, and need improvement. Please improve the manuscript in due time.

Reviewers' comments:

Reviewer's Responses to Questions

**Comments to the Author**

1. Is the manuscript technically sound, and do the data support the conclusions?

Reviewer #1: Yes

Reviewer #2: Partly

2. Has the statistical analysis been performed appropriately and rigorously? 

Reviewer #1: Yes

Reviewer #2: Yes

3. Have the authors made all data underlying the findings in their manuscript fully available?

Reviewer #1: Yes

Reviewer #2: Yes

4. Is the manuscript presented in an intelligible fashion and written in standard English?

Reviewer #1: Yes

Reviewer #2: No

5. Review Comments to the Author

Reviewer #1: - Figures 1-7 are often illegible. Authors need to make improvements so that the legend, axis etc are clearly readable.

- Authors should comment on lack of dose-dependent correlation for several extracts in free radical scavenging assay in Table 2.

- Authors should comment on the relationship/evidence behind potential presence of antioxidants and relevance as therapeutic agents.

- Authors should comment on variable response to ACEA and ACNB extracts at different doses, between 500 and 1000.

- Statement "Hence, plants with high antioxidant capacity have a notably beneficial impact on human well-being and physical condition." on line 320 is not justified.

- Results in Table 6 are somewhat surprising and contradictory to results presented earlier. ACEA extracts did not demonstrate any efficacy at dose lower than 1000 mg, however, there seems to be a correlation to weight gain even at dose as low as 250 mg. On the other hand, ACNB extras were shown to be effective at even lower doses on the other experiments, but only the 1000 mg really leads to weight gain. Could authors please comment on this?

Reviewer #2: Its a comprehensive study, which is appreciable.

But, the study does not provide detailed information on the specific molecular mechanisms by which the Atriplex crassifolia extracts exert their antidiabetic effects. While it mentions the presence of compounds like myricetin, quercetin, and sinapic acid, which are known to have antidiabetic properties, it doesn't elucidate how these compounds specifically act in this context. The research fails to identify or investigate specific drug targets. There's no mention of receptor binding studies, or other experiments that would pinpoint the exact molecular targets of the active compounds in the extracts. Moreover, this study doesn't explore the potential interactions between the various compounds found in the extracts, nor does it investigate how these compounds might interact with known antidiabetic drug targets or pathways. These are indeed significant limitations of the study.

I suggest please report the the drug target along with proper Molecular mechanism, In-silico studies (Molecular Docking, Molecular Dynamics Simulation or Machine learning and SAR of the said compounds) based studies could be used or invitro studies could be also used; but ultimately a proper mechanism of action should be established.

6. PLOS authors have the option to publish the peer review history of their article (what does this mean? ). If published, this will include your full peer review and any attached files.

**Do you want your identity to be public for this peer review?** For information about this choice, including consent withdrawal, please see our Privacy Policy .

Reviewer #1: No

Reviewer #2: **Yes: ** Abhijit Debnath

---

## [Author Response · Author response to Decision Letter 1]

17 Jun 2025

Response Sheet

[PONE-D-24-40010] - [EMID:0053111b80385d9c]

We appreciate you giving us the chance to make revisions to our manuscript, " Investigation of phytochemical and biochemical attributes of hypoglycemic activity of Atriplex crassifolia (C.A. Mey) extracts in alloxan diabetic animal model" Your insightful and beneficial comments have helped us to improve our manuscript. Every comment has been thoroughly reviewed, and appropriate modifications have been made. We trust that all the modifications have been carried out correctly, and we hope to receive approval. All the corrections and responses to reviewer comments are as follows.

Editor Comments

The manuscript entitled “Investigation of phytochemical and biochemical attributes of hypoglycemic activity of Atriplex crassifolia (C.A. Mey) extracts in alloxan diabetic animal model” is interesting although poorly written and presentation is very poor

Thank you for your comment. We have tried to improve the writing and presentation of manuscript

The antioxidant activity of extract was evaluated with highest dose 1000 µg/ml; how the IC50 values of n-hexane extract, and dichloromethane extract of A. crassifolia was evaluated, even higher than the highest concentration of experimentation?

Thank you for your comment. IC50 value represents half of the maximal inhibitory concentrations against free radicals. n-hexane and dichloromethane displayed higher IC50 values even at the highest concentration of 1000 µg/ml so they could not reach the 50% inhibition that shows they have less antioxidant potential.

The figures in the manuscript are very poor quality, and need improvement. Please improve the manuscript in due time.

We have tried to improve figures.

Reviewer 1

Figures 1-7 are often illegible. Authors need to make improvements so that the legend, axis etc are clearly readable.

Thank you for your suggestion. We have tried to improve the figures as suggested.

Authors should comment on lack of dose-dependent correlation for several extracts in free radical scavenging assay in Table 2.

Thank you for your comment. We acknowledge the lack of dose dependent correlation with various extracts in DPPH free radical scavenging assay. This variability in antioxidant potential may be attributed to the difference in the composition of bioactive compounds in the tested extracts. The non-linear relationship might be due to effectiveness of certain phytoconstituents at specific concentrations. Scavenging activity of certain constituents plateaus beyond certain concentration (ceiling effect) that leads to little or no response at particular concentrations (Huang et al., 2005).

Huang, D., Ou, B., & Prior, R. L. (2005). The chemistry behind antioxidant capacity assays. Journal of Agricultural and Food Chemistry, 53(6), 1841-1856. https://doi.org/10.1021/jf030723c

Authors should comment on the relationship/evidence behind potential presence of antioxidants and relevance as therapeutic agents.

Thank you for your suggestion. We have added the therapeutic relevance of antioxidants in Lines 325-331 as follows

The therapeutic activity of most of the plants is attributed to their potent antioxidant capability. Free radicals play a vital role in pathogenesis of various diseases. Phytoconstituents particularly phenols and flavonoids serve as the best exogenous antioxidants. They not only prohibit ROS, but also boost the endogenous defense mechanism of the body against free radicals. Phenols and flavonoid containing plants exhibit a wider range of biological activities than the conventional hydrogen donating agents. In our study, the antioxidant capabilities of A.crassifolia may be responsible for its significant antidiabetic effects.

Authors should comment on variable response to ACEA and ACNB extracts at different doses, between 500 and 1000.

Thank you for your valuable suggestion. The non-linear dose-response relationship of the extracts might be due to altered therapeutic behavior of the phytoconstituents. At higher doses, receptor saturation or downstream signaling may result in non-proportional therapeutic efficacy. Receptor binding limitations might lead to a plateau in glucose lowering capability. Moreover, the bioavailability of the bioactive compounds might decrease at higher concentrations due to enhanced metabolism. The plant constituents exhibit both synergistic and antagonistic properties at various concentrations. If the antagonistic behavior predominates at higher concentrations, this may lead to a variable dose-response relationship.

Efferth T, Koch E. Complex interactions between phytochemicals. The multi-target therapeutic concept of phytotherapy. Current drug targets. 2011 Jan 1;12(1):122-32.

Statement "Hence, plants with high antioxidant capacity have a notably beneficial impact on human well-being and physical condition." on line 320 is not justified.

Thank you for your comment. We have justified the statement as follows

Hence, plants with high antioxidant capacity have a notably beneficial impact on human well-being and physical condition. This is due to their enhanced therapeutic efficacy due to the presence of antioxidant phytoconstituents as discussed above.

Results in Table 6 are somewhat surprising and contradictory to results presented earlier. ACEA extracts did not demonstrate any efficacy at dose lower than 1000 mg, however, there seems to be a correlation to weight gain even at dose as low as 250 mg. On the other hand, ACNB extract were shown to be effective at even lower doses on the other experiments, but only the 1000 mg really leads to weight gain. Could authors please comment on this?

Thank you for your valuable comment. ACEA did not show hypoglycemic effect at dose lower than 1000 mg, however, it displayed a weight gain at dose as low as 250 mg. It might be proposed that the mechanisms of weight modulation and antidiabetic activity might not be aligned, and the weight gain was probably due to some other underlying mechanism. ACEA might contain compounds that cause wight gain independently through some metabolic effects that are not linked with glucose regulation. Conversely, ACNB displayed hypoglycemic activity at the lowest dose of 250 mg but caused weight gain at 1000 mg. it might contain constituents that cause hypoglycemic affect primarily without affecting weight and at high doses, additional metabolic effects might cause weight gain. This needs to be further evaluated through studies focused on appetite regulation, lipid metabolism and insulin sensitivity to figure out the exact mechanism behind weight changes.

Reviewer 2

Its a comprehensive study, which is appreciable.

We appreciate your valuable comment

The study does not provide detailed information on the specific molecular mechanisms by which the Atriplex crassifolia extracts exert their antidiabetic effects. While it mentions the presence of compounds like myricetin, quercetin, and sinapic acid, which are known to have antidiabetic properties, it doesn't elucidate how these compounds specifically act in this context. The research fails to identify or investigate specific drug targets. There's no mention of receptor binding studies, or other experiments that would pinpoint the exact molecular targets of the active compounds in the extracts. Moreover, this study doesn't explore the potential interactions between the various compounds found in the extracts, nor does it investigate how these compounds might interact with known antidiabetic drug targets or pathways. These are indeed significant limitations of the study.

Thank you for your valuable comment. We have included the limitations in Lines 419- 425.

The hypoglycemic mechanism of myricetin, quercetin and sinapic acid have been explained between lines (264-275). We appreciate the insightful comments of the reviewer regarding the drug targets and the exact molecular mechanisms underlying the hypoglycemic potential of Atriplex crassifolia extracts. The primary focus of the current study was to explore the hypoglycemic effect of Atriplex crassifolia extracts and the possible bioactive constituents responsible for it. Quercetin, myricetin and sinapic acid are well known antidiabetic compounds and they exert hypoglycemic effect through various mechanisms as stated in text (lines 264-275). However, further studies are required to understand the exact molecular targets as well as pathway analysis for a more comprehensive study. We also acknowledge another limitation of the study regarding the potential interactions between various compounds found in extracts and the synergistic effects of active constituents of Atriplex crassifolia with other antidiabetic agents. Our study primarily serves as the initial step in phytochemical constituents profiling and establishing their therapeutic relevance with the aim of contributing some valuable preliminary data and supporting the need to further explore the plant constituents. We have added the limitations of our study in the discussion section and proposed the future recommendations for advanced studies on Atriplex crassifolia.

---

## [Decision Letter · Decision Letter 1]

13 Jul 2025

PONE-D-24-40010R1Investigation of phytochemical and biochemical attributes of hypoglycemic activity of Atriplex crassifolia (C.A. Mey) extracts in alloxan diabetic animal modelPLOS ONE

Dear Dr. Hafeez Kamran,

Thank you for submitting your manuscript to PLOS ONE. After careful consideration, we feel that it has merit but does not fully meet PLOS ONE’s publication criteria as it currently stands. Therefore, we invite you to submit a revised version of the manuscript that addresses the points raised during the review process.

We look forward to receiving your revised manuscript.

Kind regards,

Rajesh Kumar Singh, Ph.D.

Academic Editor

PLOS ONE

Journal Requirements:

Reviewers' comments:

Reviewer's Responses to Questions

**Comments to the Author**

1. If the authors have adequately addressed your comments raised in a previous round of review and you feel that this manuscript is now acceptable for publication, you may indicate that here to bypass the “Comments to the Author” section, enter your conflict of interest statement in the “Confidential to Editor” section, and submit your "Accept" recommendation.

Reviewer #2: All comments have been addressed

Reviewer #3: (No Response)

2. Is the manuscript technically sound, and do the data support the conclusions?

Reviewer #2: Yes

Reviewer #3: Yes

3. Has the statistical analysis been performed appropriately and rigorously? 

Reviewer #2: Yes

Reviewer #3: Yes

4. Have the authors made all data underlying the findings in their manuscript fully available?

Reviewer #2: No

Reviewer #3: Yes

5. Is the manuscript presented in an intelligible fashion and written in standard English?

Reviewer #2: No

Reviewer #3: Yes

6. Review Comments to the Author

Reviewer #2: I have found that the revised manuscript is updated and all the comments has been addressed correctly.

Reviewer #3: 1. The figures are still illegible. The authors need to make improvements to make them clear and readable. Mention the improvements made with the figures to meet the standards of the journal.

2. Do the authors consider alternative analyses for dose-dependent responses?

3. The explanation regarding the compounds, although acceptable, still seems vague. Elaborate or use a graphical representation.

4. The response to the comment on the statement on antioxidants improving well-being is generic. Provide clinical evidence.

5. There are issues with the language used. Use concise scientific language.

7. PLOS authors have the option to publish the peer review history of their article (what does this mean? ). If published, this will include your full peer review and any attached files.

**Do you want your identity to be public for this peer review?** For information about this choice, including consent withdrawal, please see our Privacy Policy .

Reviewer #2: No

Reviewer #3: No

---

## [Author Response · Author response to Decision Letter 2]

7 Sep 2025

Reviewer #3: 1. The figures are still illegible. The authors need to make improvements to make them clear and readable. Mention the improvements made with the figures to meet the standards of the journal.

Thank you for your valuable comment. We have attempted to enhance the figures by increasing their dpi to comply with the journal's specifications.

2. Do the authors consider alternative analyses for dose-dependent responses?

We thank the reviewer for the valuable suggestion. Our current study design aimed at evaluating the hypoglycemic activity of A.crassifolia extracts at various doses. As per your comment, yes, dose-dependent analysis would provide additional insights, Future studies might be carried out to observe a potential dose-related effect, which we have now acknowledged and discussed in the revised manuscript under the “Conclusion” section.

3. The explanation regarding the compounds, although acceptable, still seems vague. Elaborate or use a graphical representation.

I appreciate your insightful remark. We have attempted to elucidate the function of compounds and rephrased certain sentences to enhance clarity. We expanded the text but did not include any graphical representations, as the manuscript already contains eight figures.

4. The response to the comment on the statement on antioxidants improving well-being is generic. Provide clinical evidence.

We are thankful to reviewer for the valuable suggestion. We have provided clinical evidence of antioxidants improving well-being with reference. We have added the following statement

A crucial aspect of general well-being is the improvement of metabolic health. Meta-analysis of randomized controlled trials has substantiated clinical evidence that antioxidant supplementation in obese persons significantly lowers fasting blood glucose levels, enhances insulin resistance, and diminishes lipid peroxidation.

5. There are issues with the language used. Use concise scientific language.

Thank you for your valuable comment. We have tried to review the manuscript and rectify the scientific language where needed.

---

## [Editor Report · Decision Letter 2]

14 Sep 2025

Investigation of phytochemical and biochemical attributes of hypoglycemic activity of Atriplex crassifolia (C.A. Mey) extracts in alloxan diabetic animal model

PONE-D-24-40010R2

Dear Dr. Hafeez Kamran,

We’re pleased to inform you that your manuscript has been judged scientifically suitable for publication and will be formally accepted for publication once it meets all outstanding technical requirements.

Kind regards,

Rajesh Kumar Singh, Ph.D.

Academic Editor

PLOS ONE
---

## [Editor Report · Acceptance letter]

PONE-D-24-40010R2

PLOS ONE

Dear Dr. Hafeez Kamran,

I'm pleased to inform you that your manuscript has been deemed suitable for publication in PLOS ONE. Congratulations! Your manuscript is now being handed over to our production team.

Kind regards,

on behalf of

Dr. Rajesh Kumar Singh

Academic Editor

PLOS ONE